# Multi-Strain and -Species Investigation of Volatile Metabolites Emitted from Planktonic and Biofilm *Candida* Cultures

**DOI:** 10.3390/metabo12050432

**Published:** 2022-05-11

**Authors:** Shane Fitzgerald, Ciara Furlong, Linda Holland, Aoife Morrin

**Affiliations:** 1Insight Science Foundation Ireland Research Centre for Data Analytics, National Centre for Sensor Research, School of Chemical Sciences, Dublin City University, Dublin 9, Ireland; shane.fitzgerald28@mail.dcu.ie; 2School of Biotechnology, Dublin City University, Dublin 9, Ireland; ciara.furlong6@mail.dcu.ie (C.F.); linda.holland@dcu.ie (L.H.)

**Keywords:** volatilomics, *Candida*, biofilm, fungi, pathogen, gas chromatography, volatile metabolites, solid phase microextraction, mass spectrometry

## Abstract

*Candida parapsiliosis* is a prevalent neonatal pathogen that attains its virulence through its strain-specific ability to form biofilms. The use of volatilomics, the profiling of volatile metabolites from microbes is a non-invasive, simple way to identify and classify microbes; it has shown great potential for pathogen identification. Although *C. parapsiliosis* is one of the most common clinical fungal pathogens, its volatilome has never been characterised. In this study, planktonic volatilomes of ten clinical strains of *C. parapsilosis* were analysed, along with a single strain of *Candida albicans.* Headspace-solid-phase microextraction coupled with gas chromatography-mass spectrometry were employed to analyse the samples. Species-, strain-, and media- influences on the fungal volatilomes were investigated. Twenty-four unique metabolites from the examined *Candida* spp. (22 from *C. albicans*; 18 from *C*. *parapsilosis*) were included in this study. Chemical classes detected across the samples included alcohols, fatty acid esters, acetates, thiols, sesquiterpenes, and nitrogen-containing compounds. *C. albicans* volatilomes were most clearly discriminated from *C. parapsilosis* based on the detection of unique sesquiterpene compounds. The effect of biofilm formation on the *C. parapsilosis* volatilomes was investigated for the first time by comparing volatilomes of a biofilm-positive strain and a biofilm-negative strain over time (0–48 h) using a novel sampling approach. Volatilomic shifts in the profiles of alcohols, ketones, acids, and acetates were observed specifically in the biofilm-forming samples and attributed to biofilm maturation. This study highlights species-specificity of *Candida* volatilomes, and also marks the clinical potential for volatilomics for non-invasively detecting fungal pathogens. Additionally, the range of biofilm-specificity across microbial volatilomes is potentially far-reaching, and therefore characterising these volatilomic changes in pathogenic fungal and bacterial biofilms could lead to novel opportunities for detecting severe infections early.

## 1. Introduction

*Candida parapsilosis* is an emerging pathogen that typically resides as a human commensal with limited pathogenicity. However, *C. parapsilosis* has been highlighted as a growing infectious burden due to a growing prevalence of infections arising from blood and indwelling medical devices [1], as well as its growing prevalence in neonatal sepsis [2]. *C. parapsilosis* also has a strain-specific ability to produce biofilms that determine differences in human pathogenicity across the species [3]. *Candida albicans* is a more clinically prevalent organism within the *Candida* genus. It is a polymorphic fungus that can grow either as ovoid-shaped budding yeast; as an elongated ellipsoid cell with pseudohyphae; or as a parallel-walled true hyphal form which is associated with increased virulence. This polymorphism allows *C. albicans* to infect a wide variety of host niches as it shifts its metabolism, expression of adhesions, ability to form biofilm, and virulence with each morphological transition. Morphological transitions are mediated by environmental cues [4], such as temperature, pH, O_2_ and CO_2_ content, quorum sensing interactions with neighbouring microbes, and the emission of volatile signalling compounds [5]. As the fungus transitions from the yeast cell to the ellipsoid cell to the true hyphal form, its virulence increases—this is also reversible. In contrast to *C. albicans*, *C. parapsilosis* cannot form true hyphae, it instead physiologically exists as an ovoid-shaped budding yeast or pseudohyphal form [1]. As a result, *C. parapsilosis* infections are typically less diverse and less severe than *C. albicans* infections.

Microorganisms have evolved an ability to utilise a wide variety of metabolic pathways to survive in constantly changing environments. These pathways include the metabolism of sugars, amino acids, fatty acids, sulfur- and nitrogen-containing compounds, and terpenes [6]. Volatile organic compounds (VOCs) are produced as byproducts at each stage of each respective pathway [7]. The species- and strain-specific ways in which microbes regulate their metabolism significantly contributes to the complexity of characterising their volatilomes. For this reason, in order to obtain comprehensive data for a specific microbe or microbial group, it is necessary to track species- and strain-level volatilomic diversity across the genus. Many metabolites are commonly emitted across microbial species [8]; however, the whole array of compounds that are emitted, and the abundances by which they are emitted are species-specific and are collectively referred to as its volatilome. The composition of volatilomes depends on multiple factors, such as nutritional substrates [8,9,10], strain-to-strain metabolic variation [11,12], growth phase of cells [13], and pH [14]. As a result, broad characterisation of microbial volatilomes is one of the major challenges of the field. Significant progress has been made in the last five years in building awareness of the field with the publishing of comprehensive reviews [6,15,16,17] and books [18]. Tackling challenges in the field will require comprehensive and standardised experimental workflows that support broad untargeted screening and identification of metabolites across a wide range of microbial species. Although there have been recent investigations of the volatilomes of several prevalent *Candida* genera [19,20], there is a need for more studies to support and validate these works. The examined *Candida* volatilomes have demonstrated low specificity in the early stages of growth, but develop discriminative features in the latter stages of cellular development. The chemical composition of these volatilomes have been shown to be rich in acids, aldehydes, alcohols, hydrocarbons, esters, terpenic compounds, sulfur-containing compounds, and phenols. *C. albicans*, *Candida tropicalis*, and *Candida glabrata* [19,20] are among the genus that have been investigated. However, despite being one of the most frequently isolated *Candida* genera in clinics, volatilomic data on *C. parapsilosis* are very limited; the only study available [21] reported the detection of just three compounds (ethanol, 2-phenylethyl alcohol, and 3-octanone).

Similarly with the majority of microbes, very little is known about how biofilm formation affects the emission of volatile compounds. Volatilomic discrimination of biofilms of wound-associated pathogens was recently demonstrated using a highly efficient open flow system that was coupled with both direct mass spectrometry and headspace-solid-phase microextraction coupled with gas chromatography-mass spectrometry (HS-SPME GCMS) to allow real-time analysis of steady-state biofilms [22]. Although this study demonstrated a comprehensive experimental workflow for investigating biofilms, the volatilomic emissions were compared to uninoculated media controls; therefore, the specificity of VOCs emitted from the biofilm itself was not determined. We hypothesise that this specificity can be determined through the dual investigation of biofilm-positive and biofilm-negative strains of a respective microbe. We also hypothesise that biofilm development will cause kinetic shifts in the volatilome of the biofilm-forming microbe. Following the production of biofilm, bacteria and fungi slow their metabolism to regulate the use of available substrates. This allows them to survive in challenging environments, and can also render them more tolerant to antimicrobial drugs and stimuli as the reduction in central metabolic flux reduces the intake of toxins [23]. Currently, it has yet to be investigated whether this reduction in central metabolic activity during biofilm development has measurable effects on the volatilome of microbes.

In this research, we obtain comprehensive multi-strain volatilomes for planktonic *C. parapsilosis* (10 different strains) using a standardised HS-SPME GC-MS analysis workflow [11,13] and compare it to the volatilome of a *C. albicans* strain obtained under the same conditions to understand species- and strain-specific differences across the genus. Comprehensive volatilomic data are reported across varying growth conditions for *C. parapsilosis*, an understudied clinical pathogen. Furthermore, we also apply our workflow in a volatilomic study of *Candida* biofilms for the first time to identify metabolites emitted in biofilm-specific metabolic pathways, by comparing biofilm-positive and biofilm-negative strains of *C. parapsilosis.* This final aspect of our research makes a significant contribution to the study of biofilm volatilomics by demonstrating specificity, and highlights new opportunities for further study in this research area.

## 2. Results

### 2.1. Discriminative Volatilomics of Planktonic Candida spp. at the Species- and Strain-Level

The heatmap shown in Figure 1 clearly illustrates the species-level discrimination of the volatilomes of one strain of *C. albicans* and 10 clinical strains of *C. parapsilosis* cultured in YPD media. Hierarchical clustering is a bottom-up unsupervised learning method that characterises samples within a data set based on their similarity to each other. Each sample initially represents its own cluster, and is then subsequently clustered to similar clusters until all of the samples fall under one large cluster; this large complex cluster is called a dendrogram. The individual arms of the dendrogram represent the clusters, and the length of each arm represents the Euclidean distance or dissimilarity between the samples. Therefore, the longer the arm of one cluster, the more dissimilar it is from the rest of the samples. The columns of the heatmap represent the mean abundance values of the single strain of *C. albicans* (*n* = 5) and the 10 *C. parapsilosis* clinical strains (*n* = 3). Autoscaling [24] of the abundance values was employed as the normalisation technique for this analysis. Autoscaling normalises the abundance values for each emitted compound with respect to their occurrence across the fungal samples. Although it is an effective normalisation method, it is limited by amplifying variances in the data due to inflation of extremely low and high abundance values. Following background subtraction of the whole sample volatilomes, a total of 25 unique compounds were recovered from the broad volatilomic screening of the *C. albicans* and *C. parapsilosis* isolates. Out of these 25 compounds, 23 were emitted by *C. albicans*, and 18 were emitted by *C. parapsilosis*. Therefore, on a qualitative level the volatilomes of these two fungal pathogens were highly similar. However, there were large differences in compound abundances recovered from *C. albicans* and *C. parapsilosis*; as for most compounds, *C. albicans* emitted higher abundances than *C. parapsilosis*. This occurrence, coupled with the fact that *C. albicans* emitted a greater number of compounds than *C. parapsilosis*, would indicate that it is potentially more metabolically active than *C. parapsilosis*. In Figure 1, minor strain-level variation can be observed as CP6; CP4 and CP9 are not clustered with the other CP strains. These strains are discriminated from the other strains since they emit relatively higher abundances of several compounds. Overlayed and individual chromatograms for *C. parapsilosis* and *C. albicans* volatilomes are shown in Appendix A. The species-level and strain-level diversities of the *Candida* spp. volatilomes will be further discussed with respect to the individual unique compounds in the next section.

### 2.2. Chemical Composition of Planktonic Candida Volatilomes

Extracted *Candida* volatilomes consisted of alcohols, fatty acid esters, pyrroles, ketones, and acetates. The boxplots shown in Figure 2 show the abundance of each compound recovered from the *C. albicans* samples and the *C. parapsilosis* clinical strains. The *C. parapsilosis* volatilome was primarily characterised by significant abundances of ethanol; 1-butanol, 3-methyl-; 1-butanol, 3-methyl-, acetate; and phenylethyl alcohol. Other highly abundant compounds included the following fatty acid esters: butanoic acid, ethyl ester; propanoic acid, ethyl ester; and propanoic acid, 2-methyl, ethyl ester. These compounds were also highly abundant in the *C. albicans* volatilome, that of which also contained propanoic acid, pentyl ester. Several sesquiterpene compounds were emitted by the *C. albicans* samples; these were farnesol, nerolidol, trans-farnesol, and 2,3-dihydrofarnesol. These compounds were emitted in relatively moderate to high abundances in these samples, and significantly contributed to the discrimination of the *C. albicans* volatilome from the *C. parapsilosis* volatilome. Another notable difference between the two fungal volatilomes was the emission of the ketone acetoin, which was highly abundant in all *C. parapsilosis* samples, and significantly less than in the *C. albicans* samples. Methyl thiolacetate is a sulfur-containing compound that was only recovered in *C. albicans* samples. Within the 10 clinical strains of *C. parapsilosis*, there were varying emissions of certain compounds, such as ethyl acetate, phenylethyl alcohol, propanoic acid, 2-methyl, ethyl ester, and 1-butanol, 3-methyl-, acetate. Succinimide and 3-buten-1-ol, 3-methyl- were only recovered from specific *C. parapsilosis* strains and were absent from *C. albicans* samples. These results highlight notable differences across species, as well as a high degree of stability across the *C. parapsilosis* volatilome at the strain-level.

### 2.3. Media-Dependent Influences on Planktonic Candida Volatilomes

Significant differences were observed in the compositions of the planktonic *Candida* volatilomes when the culture media were changed from YPD to TSB. CP1 and *C. albicans* were investigated to assess how stable their volatilomes were across the different nutritional media. When cultured in TSB media at 37 °C, following background subtraction, only 12 unique compounds were recovered from *C. albicans* samples (compared to 22 in YPD), and seven unique compounds were recovered from the CP1 samples (compared to 18 in YPD). The grouped bar charts shown in Figure 3 visualise the compositional influence that varying media had on the *Candida* volatilomes. In the case of the *C. albicans*, 11 out of 12 of the compounds recovered from the TSB were also present in its volatilome when cultured in YPD, with the exception being 1-pentanol, 2-methyl-. In the case of the CP1 TSB samples, six of the seven compounds that were recovered from samples were also recovered from the YPD samples, with very low abundances of benzeneacetaldehyde being the exception. For both *C. albicans* and CP1, relatively high emissions of ethanol; 1-butanol, 3-methyl-; and phenylethyl alcohol persisted across the two examined media. However, significantly less 1-propanol, 2-methyl and acetoin were recovered from CP1 TSB samples than from CP1 YPD samples, and similarly, significantly less of 1-butanol, 3-methyl-, acetate; 2,3-dihydrofarnesol; methyl thiolacetate; and 1-propanol, 2-methyl were recovered from *C. albicans* TSB samples than from *C. albicans* YPD samples. The observed higher numbers of compounds recovered from the YPD samples of both *C. albicans* and CP1, in addition to the significantly higher abundances of volatile compounds emitted in YPD media, confirm that both *Candida* spp. are more metabolically active when grown in YPD.

Figure 3 (bottom) shows the background volatilome of TSB and YPD blank samples. Key differences in the volatilome of the two media included significantly higher abundances of acid and sulfide compounds in the YPD blank samples; and a higher variety of pyrazine compounds in the TSB blank samples. There was an absence of fatty acid ester compounds in both *C. albicans* and CP1 TSB samples. YPD broth contains relatively high abundances of a variety of available acids, such as acetic acid, propanoic acid, propanoic acid, butanoic acid and butanoic acid, 3-methyl (isovaleric acid) (Figure 3, bottom), which can act as the precursors for the esterification and formation of butanoic acid, ethyl ester; propanoic acid, ethyl ester; and propanoic acid, 2-methyl, ethyl ester. The relatively high abundances of acetic acid in YPD are also responsible for the significant abundances of 1-butanol, 3-methyl-, acetate present in all *Candida* samples in YPD. This compound was likely formed by the esterification of acetic acid with 1-butanol, 3-methyl-. Low abundances of 1-butanol, 3-methyl-, acetate were recovered from the *C. albicans* TSB samples, and it was completely absent in the CP1 TSB samples (Figure 3). Similarly, the absence of ethyl acetate in the TSB samples of both *C. albicans* and CP1 indicate that a high abundance of acetic acid may be required to form ethyl acetate via esterification with ethanol. Overall, from the volatilomes obtained from the blank media (Figure 3, bottom) and the observed differences in *Candida* volatilomes across these media (Figure 3, top and middle), YPD provides an environment richer in substrates and volatilomic precursors that yield *Candida* volatilomes with higher degrees of complexity.

### 2.4. Biofilm-Dependent Influences on Candida Volatilomes

The effects of biofilm formation and maturity on *Candida* volatile emissions were examined using the sampling set up shown in Appendix A. Out of the *C. parapsilosis* strains, CP6 was chosen to be investigated due to its strong ability to produce biofilms. This is clearly shown by the crystal violet assay of the CP strains in Appendix A. Biofilm-negative CP1 (Appendix A) was analysed as a control in this investigation, as it shares a highly similar planktonic volatilome to CP6 but is unable to form a biofilm in YPD. Biofilm-negative *C. albicans* (Appendix A) was also analysed as a relative comparison to the CP samples. In order to carry out sampling of biofilm volatilomes, the experimental set-up for planktonic cultures needed to be adopted for monitoring and characterising the volatilome of maturing biofilms (see Appendix A). Volatile emissions at 24 and 48 h of biofilm growth were sampled and analysed. Unbound cells and exhausted media were washed away following sample collection at 24 h, and fresh YPD was introduced to sustain biofilm growth between 24–48 h. Although biofilm formation was not observed in the CP1 and CA samples, there was a degree of cell adhesion to the wells following PBS washes at 24 h as the cell cultures regenerated between 24–48 h. This was clearly indicated by the turbid appearance and the volatilome of the samples at 48 h. Individual compound abundances over 48 h for biofilm-negative *C. albicans* and CP1 are shown in Appendix A (*C. albicans*) and Appendix A (CP1). The comparative boxplots in Figure 4 illustrate the emissions of key compounds from the biofilm-positive CP6 strain and the biofilm-negative CP1 strain. There were clear significant differences (Appendix A) between CP1 and CP6 in the emissions of primary metabolites, such as ethanol (*p* < 0.01), 3-methyl-1-butanol (*p* < 0.01), 2-methyl-1-propanol (*p* < 0.01), and acetoin (*p* < 0.001). In biofilm-positive CP6 samples, between 24–48 h, the following short chain fatty acids were completely consumed across all samples: propanoic acid, 2-methyl propanoic acid, butanoic acid, and 3-methyl butanoic acid (Figure 5). Significant decreases (*p* < 0.01) in the abundances of acetic acid were also observed at 48 h across the biofilm-positive CP6 samples. Small abundances of butanoic acid, 2-methyl- were only detected in the CP6 samples after 48 h of biofilm growth. In biofilm-negative CP1 samples, significant decreases in acid abundances were observed in butanoic acid, propanoic acid, and 2-methyl propanoic acid (Appendix A). No significant differences in acid abundance were observed in biofilm-negative *C. albicans* cultures (Appendix A). Significant increases (*p* < 0.01) of phenylethyl alcohol were detected in the biofilm-positive CP6 between 24 and 48 h growth. This increase was not observed in either biofilm-negative *C. albicans* (Appendix A) or CP1 (Appendix A) samples. Methyl thiolacetate was detected in biofilm-positive CP6 samples at 48 h, this compound was not detected in biofilm-negative CP1 samples at any time points, but was detected in biofilm-negative *C. albicans* (Appendix A).

## 3. Discussion

In this work, we characterised the previously understudied volatilome of the emerging fungal pathogen, *C. parapsilosis*, across varying planktonic growth parameters, and compared it to the volatilome of *C. albicans*. Our primary objective for this study was to investigate species-, media-, time-, and biofilm-dependent variations across *C. albicans* and *C. parapsilosis*. The volatilomes of 10 clinical strains of *C. parapsilosis* in addition to one strain of *C. albicans* were analysed using HS-SPME-GCMS. Clear discrimination of the dissimilarities across the planktonic *C. parapsilosis* and *C. albicans* volatilomes was achieved using hierarchical clustering (Figure 1). Among the 25 unique compounds identified across both species in YPD media, 23 compounds were recovered from *C. albicans*, and 18 compounds were recovered from both *Candida* species. The most abundant compounds were the following alcohols: ethanol; 1-butanol, 3-methyl-; and 2-phenylethyl alcohol—all of which are produced through the primary metabolism of glucose and amino acids. Ethanol is typically derived from glucose metabolism and fermentation, while 1-butanol, 3-methyl- is produced from the downstream breakdown of leucine [25]. 2-phenylethyl alcohol is produced from the degradation of aromatic amino acids [6] via the shikimate pathway [26]. These core alcohols that both *Candida* species emitted are not characteristic and are common metabolic products that are widely emitted across both bacterial and fungal kingdoms [6,11,22,27]. The emission of various fatty acid esters also contributed to all *Candida* volatilomes. These included the following acetate esters: 1-butanol, 3-methyl-, acetate- and ethyl acetate; and the following short-chain ethyl esters: butanoic acid, ethyl ester; propanoic acid, ethyl ester; and propanoic acid, 2-methyl, ethyl ester. Esterification is a common metabolic process utilised by yeasts [28]. It arises from the reaction of specific alcohols and carboxylic acids, mediated by acetyl coenzyme A—a primary metabolic product formed from the decarboxylation of pyruvate [29].

The emission of all compounds discussed above was shown to be highly dependent on the growth media used for the culture (Figure 3). Volatilomic screening of microbes across different nutritional environments allows a more comprehensive view of the range of compounds that can be emitted. Microbial biosynthetic pathways for the volatile metabolites detected can be somewhat elucidated by investigating the variation in substrates and precursors available in different growth media. However, these pathways can only be fully elucidated using highly specialised techniques such as ^13^C labelling-metabolic fluxomic analyses [30,31]. YPD was the primary media examined here, as it is widely used for the growth of yeasts; TSB is a less complex universal media that can be used for both fungi and bacteria. Following broad analyses of all *Candida* strains in YPD, our goal was to determine potential synthetic pathways for the emitted compounds by analysing and comparing the *Candida* volatilomes from the acid-free and sulfide-limited TSB media. For *C. albicans* and *C. parapsilosis* (CP1), the alcohols mentioned above persisted in volatilomes recovered from both TSB and YPD media, while the fatty acid esters were essentially absent in the TSB recovered volatilomes. Volatilomic analysis of the blank TSB and YPD media revealed that YPD contained a variety of available fatty acids from which both *Candida* strains could utilise to produce fatty acid esters via esterification. Interestingly, bacteria such as *Staphylococcus aureus* and *Escherichia coli* have demonstrated the ability to produce fatty acid esters in fatty acid-free media such as TSB [11]; this is potentially due to the secondary metabolism of their primary acidic metabolites, such as acetic acid, propanoic acid, and butanoic acid, 3-methyl-.

The unique set of sesquiterpene compounds farnesol, 2,3-dihydrofarnesol, trans-farnesol, and nerolidol clearly discriminated the *C. albicans* from the *C. parapsilosis* volatilome after 24 h of incubation. These volatile sesquiterpenoid compounds have also previously discriminated *C. albicans* from *C. glabrata* and *C. tropicalis* volatilomes [20]. Farnesol is produced as a biproduct along the ergosterol synthesis pathway [32], and is a quorum sensing molecule critically used in *Candida* biofilm development. Its presence in the *C. albicans* samples and relative absence in the *C. parapsilosis* samples may also indirectly highlight the previously reported [5] biofilm formation differences between these species. It is primarily involved in the control of morphogenic transitions in *Candida*, promoting the transitions from a hypha-to-yeast [33] while inhibiting yeast-to-hypha transitions [34]. which is critical to *C. albicans* biofilm formation and virulence. Farnesol has also demonstrated strong antimicrobial properties showing anti-biofilm activity against several *Staphylococcal* spp. [35,36]. The volatile nature of these bioactive farnesol-like compounds emitted by *Candida* also indicates their potential role in long-distance inter-microbial interactions. This supports the idea of microbial volatilomics being used in novel volatile antibiotic screening [37]. Farnesol compounds are not exclusively produced by *C. albicans*, and have been previously reported to be produced in small abundances by various *Candida* spp. including *C. parapsilosis* [38]. Other previously reported bioactive components that were detected in the *Candida* volatilomes included ethanol [37,39,40]; phenylethyl alcohol [41,42]; 1-butanol, 3-methyl- (isoamyl alcohol) [37,39,40,41,42]; 1-butanol, 3-methyl-, acetate- (isoamyl acetate) [37,39,40]; ethyl acetate [37,39,40]; and propanoic acid, 2-methyl, ethyl ester [37,41]. However, although these compounds are highly common across the microbial kingdoms, it is suspected that only species-specific combinations and abundances of them elicit antimicrobial effects [43]. These effects are determined using volatilomic bioactivity assays, a field of study that is still in the early stages of development.

Volatilomic investigations involving microbial biofilms are currently lacking in the field. Notable studies describing systems for targeted volatilomic profiling of biofilms have been recently described by Slade et al. [22,44] using selected ion flow tube-MS. In this study, we demonstrate biofilm-specificity in *C. parapsilosis* volatilomes using a novel sampling approach via HS-SPME (see Appendix A) coupled with GC-MS analysis. However, there are limitations with this system that must be noted: (1) manual handling requirements for the transfer of liquids; (2) sampling containers are not specifically made for the task; and (3) increased risk of contamination. Although a standardised sampling container is not currently being constructed, manual handling requirements and contamination risks can be mitigated by exercising good aseptic technique procedures.

CP6 was confirmed as being biofilm-positive using a crystal violet assay of CP1–CP7 in YPD media (Appendix A). CP1 and *C. albicans* did not form biofilms in YPD, and were analysed across 48 h as comparative controls. The success of *C. albicans* biofilm formation is typically enhanced by coating the polystyrene wells with serum prior to inoculation [45] and using glucose-supplemented media [46]. Following 48 h of biofilm development in CP6 samples, significant differences in the abundances of various compounds were observed, particularly between 24 and 48 h (Figure 5). Whereas in biofilm-negative CP1 and *C. albicans*, following the washing step at 24 h, the cells that remained in the wells grew in the same planktonic manner as they did from 0–24 h (Appendix A). Biofilm maturity was characterised by the metabolism of short-chain fatty acids as propanoic acid, 2-methyl propanoic acid, butanoic acid, and 3-methyl butanoic acid, which all appeared to be completely consumed in CP6 samples after 24 h. Significant decreases in acetic acid also coincided with increases in the abundance of acetate molecules (Figure 5). Volatile acetic acid metabolism and the subsequent formation of acetate has previously been shown to enhance biofilm formation in bacterial cells [47]. In contrast, in non-biofilm-forming CP1 and *C. albicans*, acids produced in the initial stages remained relatively stable. Fermentation metabolites such as ethanol, and short-chain amino acid (leucine) metabolites such as 2-methyl-1-propanol and 3-methyl-1-butanol were significantly more abundant in biofilm-negative CP1 samples compared to the biofilm-positive CP6 samples (Appendix A). Acetoin was also detected at significantly higher abundances in the non-biofilm forming CP1 samples. This primary metabolite is directly derived from the breakdown of pyruvate [6] and is produced to neutralise the extracellular environment to prevent over-acidification of the cells [48]. The relatively high abundances of acetoin and these primary alcohols in CP1 samples indicate a higher rate of primary metabolism in these cells. In contrast to this, the observed reductions in abundances of these compounds in the CP6 biofilm samples is in agreement with the reports of the down-regulation of primary metabolic activity as the biofilm develops [49]. Interestingly, no significant differences in the abundances of these compounds were observed between the CP1 and CP6 when they were grown planktonically in the HS vials (Figure 2). Biofilm development in *Candida* spp. has also been associated with the upregulation of various amino acids [50]. In maturing CP6 biofilms, higher rates of aromatic amino acid degradation (shikimate pathway) were indicated through the detection of significant increases in phenylethyl alcohol detected at 48 h (Figure 5). Production of this molecule has also been linked to the stimulation of filamentous growth [51], and promotes biofilm formation in yeasts [50,52]. Despite the filamentous growth this molecule stimulates in yeasts, it has demonstrated inhibition of hyphal formation in *C. albicans* cells [26,53], that being a core step in *C. albicans* biofilm development [54]. Among the other compounds produced in the CP6 biofilms, one of the most notable was the sulfur-containing acetate, methyl thiolacetate, which was detected only after 48 h. This molecule was not detected in any of the planktonic *C. parapsilosis* strains (Figure 2)—highlighting a potential biofilm-specific role in its emission in *C. parapsilosis*. In contrast to this, methyl thiolacetate was detected in *C. albicans* cultures in both HS vials (Figure 2) and the 6-well system (Appendix A). Although YPD media was found to be rich in dimethyl disulfide and dimethyl trisulfide (Figure 3), sulfides were not detected at either 24 or 48 h of biofilm growth. Sulfur degradation is an integral metabolic pathway utilised by yeast cells for division [55] and sustaining growth [56].

Our novel sampling chamber HS-based experimental approach allowed biofilm-specific dynamics of microbial volatilomes to be explored using untargeted volatilomics. This technique could also be used to investigate potential biofilm-specificity in the volatilomes of other clinical fungal and bacterial pathogens. Another experimental application for this system in the future could be for volatilomic bioactivity assays to monitor volatile-mediated interactions between microbes grown in separate wells, under a shared headspace. A major advantage of this method is that it can simply be adapted for a variety of sampling and analysis platforms (i.e., SIFT-MS and proton transfer tube-MS), and therefore can be used for qualitative and quantitative investigations.

## 4. Methods

### 4.1. Growth of Candida Planktonic Samples

The following Candida strains were examined: CA: C. albicans (DSM 1386); C. parapsilosis (CP1: CLIB214; CP2: CDC317; CP3: CDC173; CP4: 711701; CP5: CDC167; CP6: J961250; CP7: CDC179; CP8: J930733; CP9: 103; CP10: J930631/1)—see Appendix A for reference list. Growth media used were the following: YPD: yeast peptone dextrose; TSB: tryptone soy broth. Each strain was streaked individually on yeast peptone dextrose (YPD) agar plates and incubated at 30 °C overnight. For each replicate, a single colony was inoculated in 4 mL of YPD or TSB broth and incubated at 37 °C overnight, with shaking at 180 rpm. Each replicate overnight culture was individually incubated in a 50-milliliter conical centrifuge tube. Each overnight culture was diluted to a total volume of 5 mL in growth media (YPD or TSB) to a cell count of approximately 10^7^ colony, forming units (CFU)/mL in the 20 mL headspace vials which were then sealed with magnetic polytetrafluoroethylene/silicone septum screw caps (Merck, Cork, Ireland). For samples cultured in YPD, samples of each CP strain (*n* = 3) and CA samples (*n* = 5) were incubated at 37 °C, with shaking for 24 h. The headspace (HS) of each sample was subsequently sampled and analysed (described below).

### 4.2. Growth of Candida Biofilm Samples

CP6 (J961250), CP1 (CLIB214), and *C. albicans* (DSM 1386) were investigated in this section of the study. For each replicate, a single colony was inoculated in 4 mL of YPD broth and incubated at 37 °C overnight, with shaking at 180 rpm. Each replicate culture was individually incubated in a 50-milliliter conical centrifuge tube overnight. The following day, each replicate was centrifuged at 4000 rpm for 4 min and washed with 1 mL of PBS twice. The cells were then diluted to an OD of 1 (10^7^ CFU/mL). In a 6-well plate, 2.5 mL of YPD was initially added to each well. Then, 2.5 mL of cells was deposited in each well to bring the total volume of each well to 5 mL (media:cells, 50:50). Note that to minimise the risk of contamination, it was important to ensure that the 6-well plate was covered in between each liquid transfer. The lid was placed on the 6-well plate, and the plate was carefully placed in the sampling container (‘Good For You’ 850 cm^3^ Borosilicate glass containers, dimensions: 19.3 cm × 13.6 cm × 6.7 cm). The lids of these containers have a snap lock with a silicone seal to enhance sterility. The lid of the 6-well plate was removed to momentarily expose the cultures (see Appendix A) before the lid of the sampling container was closed over the system. A layer of parafilm was wrapped around the lid of the sampling container. The container was then placed in a static incubator at 37 °C for 24 h.

Following SPME sampling of the HS of the sampling container (see next section) at 24 h, exhausted media and waste was manually removed from each well using an auto-pipette, and they were washed twice with 1 mL of PBS. Fresh media (5 mL) was then slowly deposited in each well, and the plate was re-sealed in the sampling container and re-incubated at 37 °C for 24 h. Again, to avoid contamination the system was covered in between liquid transferrals.

### 4.3. Crystal Violet Staining of Biofilm Samples

Following the HS sampling of the 6-well plate samples, the waste media was removed from each well before being washed with 1 mL of PBS (each well) twice. The plates were then left overnight to dry at room temperature. Following that, 1 mL of 0.4% crystal violet was added to each well and left for 15 min. The crystal violet was then removed and the wells were washed with 1 mL of PBS three times. The plates were then left to dry at room temperature.

### 4.4. HS-SPME Sampling

SPME fibers were used for sampling VOCs and consisted of 85-micrometer Carboxen/Polydimethylsiloxane Stableflex (2 cm) assemblies (Supelco Corp., Bellefonte, PA, USA). For the planktonic cultures, the SPME needle was pierced through the septum of the HS vial, and the fibre was exposed to the HS of the sample for 20 min while agitated. Following this, the fibre was retracted and the SPME assembly removed from the vial. The SPME fibre was then inserted into the GC inlet and thermally desorbed at 250 °C for 2 min for subsequent separation and detection by mass spectrometry. During the temporal analysis, the magnetic screw caps of each sample were also tightly covered with parafilm following each round of sampling, in order to minimise any loss of VOCs. For the biofilm samples, a septum was fixed to the lid of the sampling container with multiple layers of tape (see Appendix A). The SPME fiber was pierced through this septum and exposed to the headspace of sampling containers containing the 6-well plate samples for 30 min, while static in an incubator at 37 °C.

### 4.5. Gas Chromatography–Mass Spectrometry

An Agilent 6890 GC connected to an Agilent 5973 mass selective detector (Agilent Technologies, Inc., Santa Clara, CA, USA) was used for all analyses. Separations were performed on a DB-WAX column (Agilent Technologies Ireland, Cork) (30 m × 0.25 mm × 0.32 μm). The carrier gas used was helium, with a constant flow rate of 1.3 mL/min. For manual injections of SPME fibers, the system was equipped with a SPME Merlin Microseal (Merlin Instrument Company, Newark, DE, USA), and the inlet was maintained at a temperature of 250 °C. Split-less injection was used for all samples, with a gas purge being activated after 2 min. Each SPME fibre was desorbed for 2 min within a SPME inlet liner (Supelco). The initial GC oven temperature was 40 °C for 5 min, and was programmed to increase at a rate of 10 °C min^−1^ to 240 °C, with a final hold for 5 min at this temperature, giving an overall running time of 29 min. The transfer line temperature was set at 230 °C. The MS was operated at a scan range of 35–400 *m/z*, scan rate of 3.94 s^−1^, ion source temperature of 230 °C and ionising energy of 70 eV. Identification of compounds was performed using the National Institute of Standards and Technology (NIST) library (2017). NIST match factors of >70% were used. Retention index (RI) values for polar columns provided by the NIST Chemistry WebBook, SRD 69, were used to support the identification of these compounds. Any compound found to have an RI value ≤ 12 RI units of the RI values found in the NIST database were deemed acceptable matches. See Appendix A for the chromatographic retention and mass spectral validation of each compound. An external standard mixture of saturated alkanes (C_7_-C_30_; Merck, Cork, Ireland) was injected into the GC-MS under the same temperature conditions as the samples and used for RI matching. This was done by rapidly dipping an exhausted SPME needle into the mixture once and injecting it into the GC-MS. A fully functional SPME fiber was not used for this, since exposure to hexane degrades the fiber integrity.

### 4.6. Data Analysis

Agilent MassHunter Qualitative Analysis 10.0 software was used to analyse raw chromatographic data. Peak acquisition and the respective peak area data were calculated by employing the chromatogram deconvolution compound mining algorithm. Chromatographic peaks were compared using the NIST Chemistry WebBook. Peaks found to be from exogenous sources, such as the SPME fiber, glass vial, and column, were removed from the data set. Only peaks that could be accurately identified and that were detected in more than one replicate sample were included in the final peak list. R (version 1.2.5033) was used for data exploration and visualisation. Raw VOC data were standardised using scaling [24]. Scaling converts the values in the dataset into ratios relative to the difference in abundances between the VOCs, which allows each VOC to be equally represented in the subsequent data analysis. For compounds that were present in some replicate samples (of a given strain in a given media) and absent from others, these missing values were imputed as zero. For compounds that were absent from all replicates (of a given strain in a given media), these missing values remained as missing values. Hierarchical clustering and principal component analysis (PCA) were carried out on the data sets using the following R packages: ‘pheatmap’ (version: 1.0.12), ‘egg’ (version: 0.4.5), and ‘cluster’ (version: 2.1.0). For the hierarchical clustering analysis, Euclidean distance was used as the measure of (dis)similarity. Other R packages used for the graphics in this study were the following: ‘tidyverse’ (version: 1.3.1), ‘ggplot2’ (version: 3.3.5), and ‘ggfortify’ (version: 0.4.12). Statistical analyses were facilitated by the R package ‘ggpubr’, version: 0.4.0). Mean comparison *p*-values were calculated using the Wilcoxon test.

## 5. Conclusions

In this study, the volatilome of multiple strains of *C. parapsilosis* was comprehensively investigated for the first time under varying growth conditions. These results were compared to volatilomic analyses of *C. albicans* under the same conditions to assess inter-species variations within the *Candida* genus. A total of 24 unique compounds were identified, which allowed clear discrimination of the *C. parapsilosis* and *C. albicans* volatilomes from each other. Among these differences was the unique emission of sesquiterpene-type compounds (farnesol, 2,3-dihydrofarnesol, and nerolidol) by *C. albicans*. High degrees of stability were observed in the abundances of individual compounds detected across the 10 examined *C. parapsilosis* strains. However, volatilomic stability was not observed across different media, as the effects of culturing in TSB significantly reduced the diversity of both *C. parapsilosis* and *C. albicans* cells compared to YPD cultures. While primary metabolites were detected in similar abundances across the two media as a result of relatively similar abundances of available glucose, high acid and sulfide contents in YPD enabled the generation of esters, acetates, and sulfur-containing compounds that were not recovered from TSB cultures.

A novel biofilm-specificity study in the *C. parapsilosis* volatilome was also carried out. Following comparative analyses, significantly fewer primary metabolites were detected in the biofilm-positive *C. parapsilosis* samples than the biofilm-negative cultures. Furthermore, in biofilm-positive samples, significant consumption of all short-chain fatty acids was observed, while the unique increase in the abundance of phenylethyl alcohol correlated with biofilm maturity. This novel HS sampling set-up has a wide range of potential applications, from biofilm volatilomic monitoring, to characterisation of microbial co-cultures and bioactivity assays. By examining the effects of strain-to-strain variation, media- and time-dependent emission, and biofilm formation, a more comprehensive view of the metabolic capabilities of microbes can be achieved. Comprehensive profiling of microbes in this manner will ultimately allow a simpler translation of microbial volatilomics workflows into clinical volatilomic applications.

## Figures and Tables

**Figure 1 metabolites-12-00432-f001:**
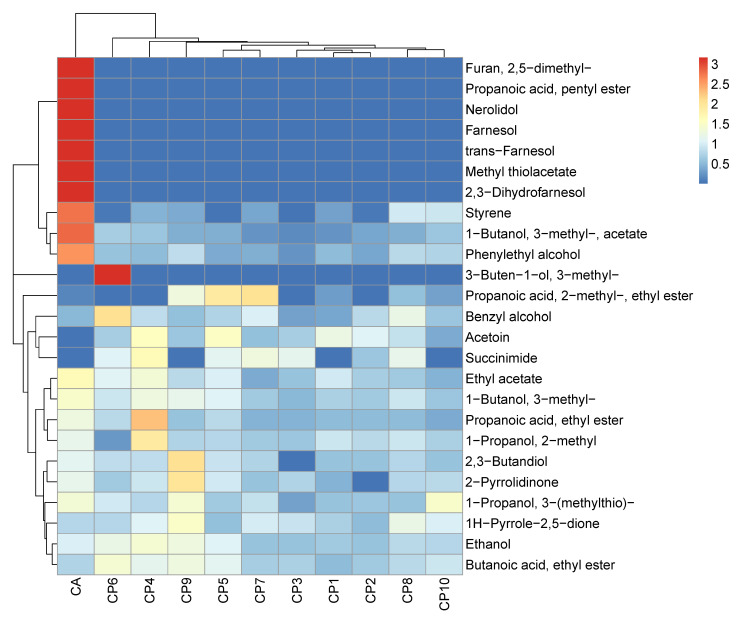
Heatmap plot illustrating hierarchical clustering characterisation of the planktonic *Candida* volatilomes in YPD media. Each column represents the mean abundance values from each examined *Candida* strain (*C. parapsilosis*, *n* = 3; *C. albicans*, *n* = 5). Abundance values were auto-scaled across each row, according to each compound.

**Figure 2 metabolites-12-00432-f002:**
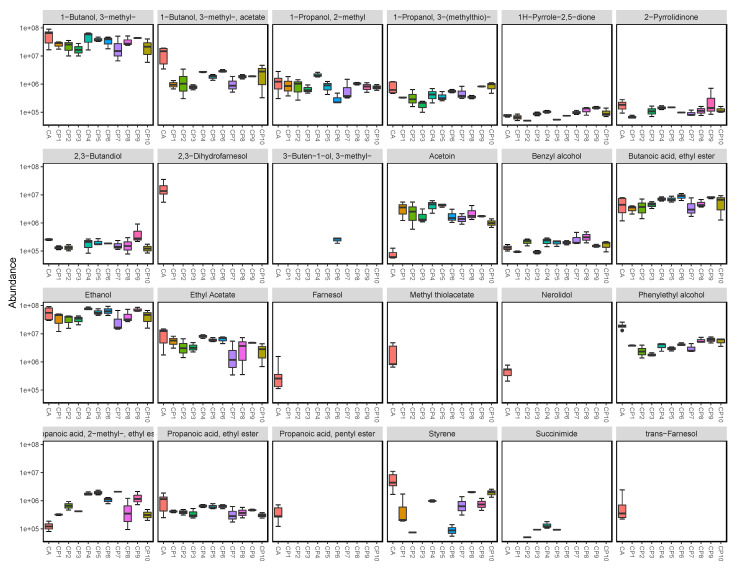
Individual comparative boxplots illustrating the emission of each compound by each of the examined planktonic *Candida* strains (CP 1–10 *n* = 3; CA, *n* = 5). The y-axis of each plot was scaled by log10 to improve visibility and interpretability of the plots. Y-axis labels are displayed in scientific notation where ae+b = a × 10^b^. Names of compounds that are not fully visible: row 4, 1st: propanoic acid, 2-methyl-,ethyl, ester.

**Figure 3 metabolites-12-00432-f003:**
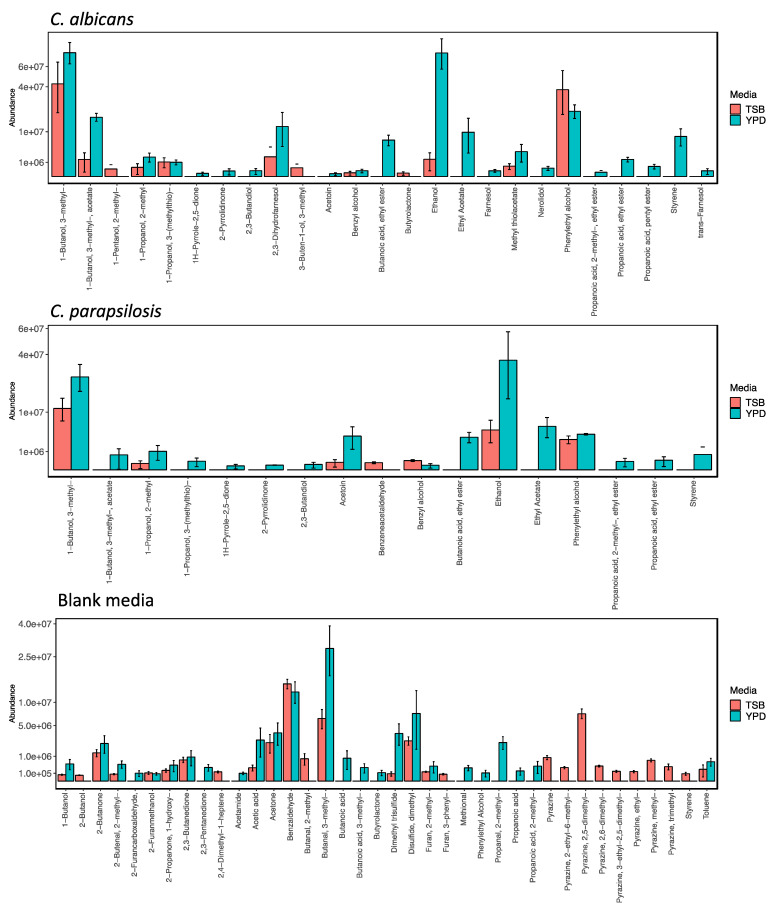
Cross-media comparative bar charts illustrating the volatilomes of *C. albicans* (**top**, *n* = 3) and *C. parapsilosis* (**middle**, *n* = 3) cultures in TSB (*n* = 5) and in YPD (*n* = 5) media, and TSB and YPD blank samples (**bottom**). Y-axis labels are displayed in scientific notation where ae+b = a × 10^b^.

**Figure 4 metabolites-12-00432-f004:**
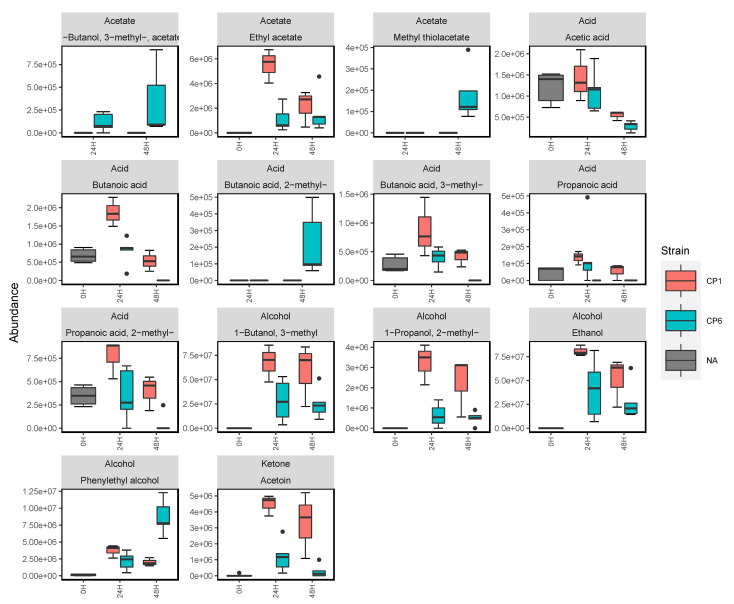
Grouped boxplots comparing the abundances of compounds emitted from biofilm-negative forming CP1 (*n* = 3) and biofilm-positive CP6 (*n* = 5). In both cases, the volatilomes were sampled and analysed using the experimental set-up shown in Appendix A. Y-axis labels are displayed in scientific notation where ae+b = a × 10^b^.

**Figure 5 metabolites-12-00432-f005:**
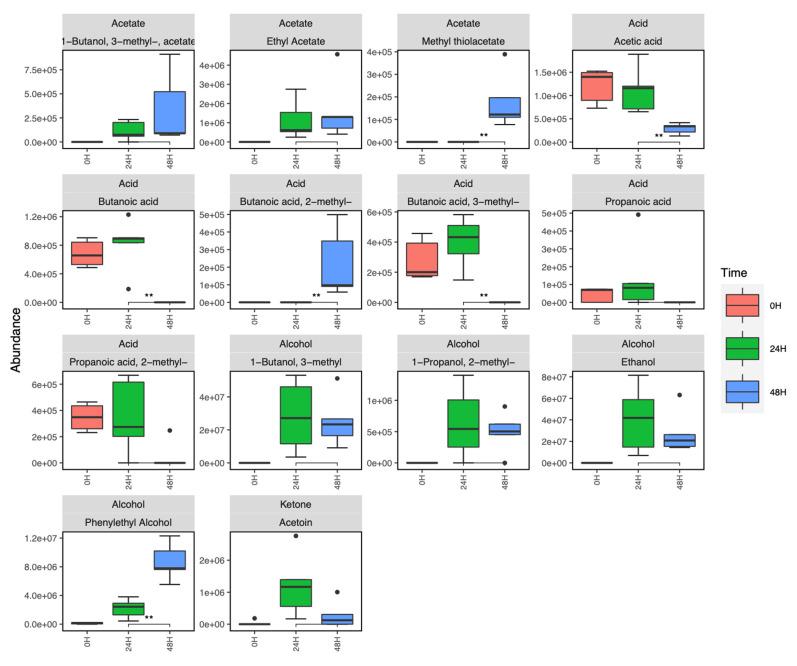
Compound boxplots illustrating the abundances of various compounds detected in the headspace of CP6 biofilm cultures at 0, 24 and 48 h of growth. Statistically significant differences in the abundance of compounds at each time point are illustrated using the star system where ** = *p* < 0.01. Y-axis labels are displayed in scientific notation where ae+b = a × 10^b^.

## Data Availability

All data used to generate the figures contained in this article can be found at: https://figshare.com/articles/dataset/Multi-Strain_and_-Species_Investigation_of_Volatile_Metaboli_tes_emitted_from_Planktonic_and_Biofilm_Candida_cultures_-_DATA/19650144 (accessed on 2 April 2022).

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
