# Peer review of "Multi-Strain and -Species Investigation of Volatile Metabolites Emitted from Planktonic and Biofilm Candida Cultures"

_metabolites, 2022, doi:10.3390/metabo12050432_

Round 1

Reviewer 1 Report

Manuscript ID: metabolites-1696776

MANUSCRIPT SUMMARY

This manuscript is a survey of the volatile compounds emitted across several strains of a Candida species, C. parapsilosis, and a C. albicans strain was included as a comparison.  Compounds were identified off the EI spectra and the RI values were also used.  C. parapsilosis is a potential pathogen, so there should be some interest in that.  Biofilm-positive and biofilm-negative strains were compared as well.  Time and media specific changes were addressed as well.

Ultimately, the negatives of this manuscript include that it is lacking is any type of experimental setup or in vivo measurements and is highly descriptive.  The positives are that while a very descriptive study it is rather comprehensive and covers several strains, so if a researcher is interested in volatile compounds in C. parapsilosis they should find it interesting.  Applications to other microbial systems for volatile analysis should also be of interest.

GENERAL COMMENTS

Please be consistent with italicizing Candida.  As a genus, although an extremely heterogeneous one, it should be italicized.  Overall editing needs to be tightened up, I see where citations did not get added and an error message was inserted.

There are some interesting suffixes being added to volatile.  I guess -omics can be added to any word now.  Candida “volatilomes” looks just like a species name.  But I don’t see where “volatilomically” has ever been used before.  I understand this is a very “volatile” manuscript, but maybe check to see if the words exist or use a better descriptor.

The heatmap for figure 1 is very useful.

A figure with the GC chromatogram is always helpful for others that use this as a reference.  This could be included in the supplemental.  A table in the text is absolutely needed that lists all the pertinent information for the compounds the authors are discussing, e.g. chemical database identifiers, retention time/ and index, molecular formula, maybe unique mass if that is identifiable using this software, groups it is significant in, etc… 

SPECIFIC COMMENTS

Line 314, I think “unstudied” is supposed to be “understudied”

Data availability has not been addressed.

Author Response

Please find below point-by-point responses to Reviewer 1 comments:

- Italics have been updated. Thank you for notifying us of this.
- Wording changes have been made where “volatilomically” had previously been used on lines
206-210 and line 280-282.
- Overlaid and individual chromatograms have been added to the Supplementary Information as suggested (Figure S1 and S2), these are introduced in the text on line 214. A table containing the GC-MS identification data has also been added to the SI (Table S2), it is introduced in line 161. The authors thank the reviewer for these suggested additions.
- Line 314 has been changed
- Data is now available on Figshare at DOI: 10.6084/m9.figshare.19650144

Reviewer 2 Report

In this manuscript was described the evaluation of the volatilome of Candida albicans and parapsilosis. Despite the manuscript seems to present some relevant results there are several issues that should be addressed:

Line 1 - Protocol? or Original article?

Line 2-3 - The title should be more specific.

Line 18 - "...18 from C. parapsilosis". I have the doubt if they were 18 or 16? in line 197 of the manuscript, the authors referred 16! 

Line 29-30 - The keywords should not be numbered.

Line  50 - byproducts instead biproducts

Line 57 - These references should be merged in order to appear as [8-10]. In addition, proceed in a similar way in the following situations in the manuscript.

Line 181 - The statistical analysis method should be described.

Figure 1 - If possible the scale should be defined only with a positive scale (because these compounds could be present in minor or higher contents).

Figure 2 - The CP10 should be presented more on the right side (instead between CP1 and CP2). In addition, in the legend, was missing the number of replicates considered in this figure.

Figure 3 - The y-axis scale is not properly presented because in the low range it was not possible to understand to which values correspond. In addition, some columns were larger (for some reason?) In addition, I think that a similar scale between figures should be presented for a better comparison! Regarding the figure legend, it was missing the number of replicates considered in this figure. In this figure, it presented the mean and SEM?

In figure 3 there is a comparison between the volatilome of the candida species and the blank medium used. If the majority of compounds already exist with only the media it seems that the importance of this work could be committed. Why it was necessary to cultivate the candida species if only the blank medium already emitted the majority of the volatile organic compounds?

Line 281 - I did not understand how this could be considered a control?  I think that this is only a relative comparison between them! The control should be considered if it was uniquely used the medium.

In addition, the rationale for selecting only the CP6 (excluding all the other CPs) from the study of biofilm also should be clarified.

Line 301 - I think that this was not correct because I observe a considerable increase in figure S5 and also an increase followed by a decrease in figure S4. Please rewrite the idea.

Figures 4 and 5 - Why did not use in each figure the same y-axis scale (enabling to have more space to put the complete name of the compounds visible)?

Line 353 - Please revise the reference used!

Line 411 - I think here you want to refer to figure 3 (instead figure 4).

Author Response

Please find below point-by-point responses to Reviewer 2 comments:

           - Line 1: has been changed to ‘Original article’

  • - Line 2-3: The authors agree with this suggestion and the title of the article has been changed.

  • - Line 18: Corrections have been made to the errors in line 18.

  • - Line 29-30: Numbers have been removed from keywords
  •  
  • Line  50 - now corrected, thank you.

  • - Line 57: Unfortunately, references could not be merged as the version of Endnote that was used to prepare this manuscript will not allow without deleting references.

  • - Line 181: Description of statistical analysis added to line 191-192

  • - Figure 1: Positive scale added to Heatmap in Figure 1…colour arrangement as a result had to be changed too.

  • - Figure 2: X axis has been corrected and number of replicates have been added to figure title.

  • - Figure 3: Y axis has been corrected to display the lower abundance values. Column size has been corrected. We do not aim to compare the volatiles of C. parapsilosis with C. albicans or the blank medium in Figure 3. The goal of this Figure is to illustrate clear differences in VOC recovery across the two media. All compounds shown in the C. albicans (top) and C. parapsilosis (middle) sections of Figure 3 (with the exception of minor abundances of phenylethyl alcohol), are unique compounds that are not found in the blank media samples.
  •  

- Line 281: Biofilm-negative CP1 was used as the primary control in this section because it has a highly similar planktonic volatilome to biofilm-positive CP6 (which can be seen in Figure 2). By comparing these two strains, biofilm-specificity in volatile emission from CP6 is clearly demonstrated. The authors do agree with the reviewer that C. albicans is a relative comparison, wording has been changed to accommodate this point on line 320.

  • Line 301: The rationale for choosing CP6 for the biofilm study is described on line 316 – 318 and is supported by a biofilm assay of CP1 – CP7 shown in Figure S5 that clearly shows CP6 to be a biofilm-positive CP strain and CP1 to be a biofilm-negative strain.

  • - Figures 4 and 5: Due to the relatively high abundances of ethanol, 1-butanol, 3-methyl and phenylethyl alcohol, a shared y-axis would reduce the visual representation of the other compounds in the plot. The authors agree that the y-axis should be displayed individually for each compound for this reason.

  • - Line 353: Point of reference has been modified.

  • - Line 411: Error regarding wrong Figure reference has been corrected, the authors thank the reviewer for pointing this out.

Round 2

Reviewer 1 Report

Manuscript is much improved

Reviewer 2 Report

In general, the comments and suggestions were taken into account and the manuscript was improved.